# Standard Visual and Ordinal Coronary Calcium Scoring on PET/CT: Agreement with Agatston Scoring and Prognostic Implications

**DOI:** 10.3390/diagnostics15232969

**Published:** 2025-11-22

**Authors:** Sehyun Pak, Hye Joo Son, Dongwoo Kim, Jung Won Moon, Yoo Na Kim, Ji Young Woo, Min-Kyung Kang, Dong-Ok Won, Suk Hyun Lee

**Affiliations:** 1Department of Medicine, Hallym University College of Medicine, Chuncheon 24252, Gangwon, Republic of Korea; marbleintheworld@gmail.com (S.P.); dongok.won@hallym.ac.kr (D.-O.W.); 2Department of Nuclear Medicine, Dankook University Medical Center, Cheonan 31116, Chungnam, Republic of Korea; neuroscience@dankook.ac.kr; 3Department of Nuclear Medicine, Hallym University Sacred Heart Hospital, Hallym University College of Medicine, Anyang-si 14068, Gyeonggi, Republic of Korea; kdwoo@hallym.or.kr; 4Department of Radiology, Hallym University Kangnam Sacred Heart Hospital, Hallym University College of Medicine, Seoul 07441, Republic of Korea; moonyard@naver.com (J.W.M.); drmarie@hanmail.net (Y.N.K.); baccas@hallym.or.kr (J.Y.W.); 5Department of Cardiology, Hallym University Kangnam Sacred Heart Hospital, Hallym University College of Medicine, Seoul 07441, Republic of Korea; homes78@hallym.or.kr; 6Department of Artificial Intelligence Convergence, Hallym University, Chuncheon 24252, Gangwon, Republic of Korea; 7Department of Population and Quantitative Health Sciences, University of Massachusetts Chan Medical School, Worcester, MA 01655, USA

**Keywords:** vascular calcification, coronary artery disease, neoplasms, positron emission tomography, computed tomography

## Abstract

**Background**: Visual assessment of coronary artery calcium (CAC) on ungated chest CT has been described previously. However, its reliability and clinical utility remain uncertain, particularly in PET/CT studies that use low-dose, low-slice CT and are susceptible to respiratory artifacts. **Methods**: We retrospectively analyzed 106 patients (median age, 66 years [interquartile range, 60–75 years]; 67 men [63.2%]) who underwent PET/CT and electrocardiogram (ECG)-gated chest CT within a 90-day interval. Six readers (three radiologists and three nuclear medicine physicians) independently assessed CAC on PET/CT using a standard four-point visual scale and a 0–12 ordinal scale based solely on written instructions. Agatston scoring was also performed. Interobserver agreement and concordance with ECG-gated chest CT Agatston score categories were calculated. Major adverse cardiovascular events (MACE) were recorded over a median follow-up of 3.5 years. **Results**: Interobserver agreement was good for both the standard visual (κ = 0.761) and ordinal (κ = 0.779) scales. Concordance with ECG-gated CT Agatston categories was higher for standard visual (κ = 0.849) and ordinal (κ = 0.750) scoring than for PET/CT Agatston categories (κ = 0.464). Both qualitative scales tended to underestimate CAC categories compared with ECG-gated CT; however, severe CAC on PET/CT predicted MACE (hazard ratios: 4.41 standard visual; 6.59 ordinal), and the ordinal scale significantly stratified MACE-free survival (*p* = 0.047). **Conclusions**: Standard visual and ordinal CAC scoring on the ungated CT portion of PET/CT is quick, reproducible, closely mirrors ECG-gated-CT Agatston grading, and offers prognostic value for future MACE in cancer patients.

## 1. Introduction

Significant advancements in cancer treatment have improved survival rates, resulting in a growing population of cancer survivors who remain at increased risk for cardiovascular disease (CVD). Epidemiologic data indicate that adult cancer survivors have a 37% higher incidence of CVD than the general population, even after adjusting for traditional risk factors [1]. This elevated risk arises from shared cardiovascular and oncologic risk factors, perioperative cardiac risks associated with cancer surgery [2,3,4], cardiotoxic effects of certain cancer therapies [5], and competing cardiovascular and oncologic mortality risks [6]. As such, cardiovascular health has become a critical component of comprehensive cancer survivorship care.

[^18^F]fluorodeoxyglucose (FDG) positron emission tomography/computed tomography (PET/CT) is widely used in oncology for staging, treatment response assessment, and surveillance. Incidental findings from these scans can yield valuable health information beyond primary oncologic indications. In particular, the low-dose CT component of PET/CT often visualizes coronary arteries. Coronary artery calcification (CAC) is a well-established marker of coronary artery disease and an independent predictor of future cardiac events [7,8]. According to the Multi-Ethnic Study of Atherosclerosis (MESA), CAC is present in approximately 60% of adults overall, including 59% of Asian men and 42% of Asian women [9]. In asymptomatic individuals, elevated CAC burden is associated with a significantly increased risk of myocardial infarction and other adverse events [10]. Current clinical guidelines recommend CAC assessment using dedicated electrocardiogram (ECG)-gated CT to enhance cardiovascular risk stratification and guide preventive therapies, such as statins [11]. Nonetheless, incidental CAC on routine ungated CT is frequently underreported in clinical practice [12].

Recent guidelines from the Society of Cardiovascular Computed Tomography (SCCT) and the Society of Thoracic Radiology (STR) emphasize the importance of visually identifying and reporting CAC on all ungated chest CT scans [13,14]. Studies have demonstrated strong correlations between CAC identified on ungated chest CT and CAC quantified using gated Agatston scoring [15,16]. However, most existing data are derived from studies performed by experienced cardiothoracic radiologists [16,17,18] or by physicians who have received formal CAC scoring training [15,19]. In clinical practice, however, PET/CT scans are typically interpreted by nuclear medicine physicians who may not have dedicated cardiac training, potentially limiting reproducibility. Furthermore, the ungated, free-breathing acquisition of PET/CT can introduce motion artifacts and partial volume effects, complicating accurate CAC evaluation [20].

This study aimed to evaluate (1) the reproducibility of standard visual and ordinal CAC scoring by multiple readers, including nuclear medicine physicians, with minimal training (based solely on written instructions); (2) the concordance between various PET/CT-derived Agatston score categories and ECG-gated chest CT Agatston score categories; and (3) the prognostic value of standard visual and ordinal CAC scoring on PET/CT for predicting future cardiac events. These findings aim to inform the best practices for assessing incidental CAC on routine oncologic PET/CT.

## 2. Materials and Methods

### 2.1. Study Population

This retrospective study consecutively enrolled patients who underwent [^18^F]FDG PET/CT and ECG-gated chest CT within a 90-day interval between June 2012 and September 2023 at our institution. Patients with a history of coronary artery intervention or bypass graft surgery were excluded (Figure 1). The Institutional Review Board of our institution approved this study (IRB no. 2024-09-004), and the requirement for informed consent was waived.

Clinical data, including patient demographics, cardiovascular risk factors (hypertension, diabetes mellitus, smoking history, and dyslipidemia), current medications, and cancer type and stage, were obtained from electronic medical records (EMRs). Indications for PET/CT, most commonly for initial cancer staging or recurrence assessment, were also recorded. Major adverse cardiovascular events (MACE) during follow-up were identified by first screening EMRs using relevant cardiovascular International Classification of Diseases (ICD) codes. Potential cases were then verified through detailed EMR review. Final definitions of MACE included coronary angiography followed by revascularization, ischemic stroke, acute coronary syndrome, myocardial infarction, and new-onset high-grade atrioventricular block. Patient characteristics are summarized in Table 1.

### 2.2. Data Acquisition

All PET/CT scans were acquired using a Gemini TF 16 PET/CT scanner (Philips Healthcare, Cleveland, OH, USA). For each scan, a low-dose, non-contrast CT from the skull base to the mid-thigh was performed for attenuation correction and anatomical correlation, followed by PET acquisition using a standard protocol [21]. CT parameters included a tube voltage of 120 kVp, a tube current of approximately 50 mAs, and a slice thickness of 4 mm, reconstructed at 4 mm intervals. Collimation was 8 × 3.0 mm. Scans were acquired without ECG gating and during free breathing.

ECG-gated chest CT was performed using one of three multidetector CT scanners available during the study period: SOMATOM Definition Flash (Siemens Healthineers, Forchheim, Germany, *n* = 77), SOMATOM Drive (Siemens Healthineers, *n* = 18), or Revolution Apex (GE Healthcare, Waukesha, WI, USA, *n* = 11). All ECG-gated CT scans used standardized coronary imaging protocols with ECG gating. Tube voltage was typically 100–120 kVp, and tube current ranged from 60 to 120 mAs. Images were reconstructed as thin slices (2.5–3 mm) with overlapping intervals of 0–50%. Collimation was 64 × 0.6 mm (Definition Flash and Drive) or 256 × 0.625 mm (Revolution Apex). Table 2 summarizes PET/CT and ECG-gated chest CT acquisition parameters.

### 2.3. Coronary Artery Calcium Analysis

Agatston scores were obtained from both the ECG-gated chest CT and the ungated PET/CT using dedicated software (SmartScore 4.0; GE Healthcare, Chicago, IL, USA) on an advanced workstation (Advantage Workstation 4.7; GE Healthcare, USA) by the corresponding author, who was blinded to any clinical information during the analysis. Calcium burden was quantified separately in the left main, left anterior descending, left circumflex, and right coronary arteries. Scores were categorized as none (0), mild (1–99), moderate (100–399), and severe (≥400), with ECG-gated CT Agatston scores serving as the reference standard [22,23,24].

CAC on PET/CT was assessed using three methods: (1) Agatston score categories, (2) a standard four-point visual scale, and (3) an ordinal scale. The same numerical cut-offs were applied to PET/CT Agatston scores. For standard visual and ordinal scoring, six readers independently reviewed the PET/CT scans: three radiology residents and three board-certified nuclear medicine physicians. Readers were provided only with written instructions for CAC evaluation (Appendix A); no interactive training or feedback was given.

We used the same Agatston score categories on PET/CT as on ECG-gated CT. The standard visual scale paralleled Agatston categories, classifying each case as none, mild, moderate, or severe (Appendix A). For ordinal scoring, calcification in each of the four major coronary arteries was graded from 0 to 3 (0, no calcification; 1, <25% of the arterial length; 2, 25% to <50%; 3, ≥50%; Figure 2). Scores from all arteries were summed (range, 0–12) and recategorized as none (0), mild (1–3), moderate (4–6), or severe (7–12) (Appendix A). Although the ordinal scheme was adapted from prior studies [16,17,18], thresholds of 25% and 50% were used—instead of one-third and two-thirds—to increase sensitivity, as in the method of Choi et al. [19]. When reader assessments differed, the majority classification was used. If two or more categories were equally represented, the median category was assigned as the final score.

### 2.4. Statistical Analysis

Continuous variables were reported as medians with interquartile ranges (IQRs), and categorical variables as frequencies and percentages. Generalized Fleiss’ kappa with linear weights was calculated to assess interobserver agreement among the six readers for the standard visual and ordinal PET/CT scales. Agreement between ECG-gated chest CT Agatston score categories and the three PET/CT scales (Agatston score categories, standard visual scale, and ordinal scale) was also evaluated using generalized Fleiss’ kappa. MACE-free survival according to standard visual and ordinal PET/CT scores was assessed using Kaplan–Meier estimates and compared with the log-rank test. The Cox proportional hazards model was used to evaluate the association between CAC and MACE. Two-sided *p*-values < 0.05 were considered statistically significant. All statistical analyses were performed using R software version 4.3.0 (R Foundation for Statistical Computing, Vienna, Austria).

## 3. Results

### 3.1. Patient Characteristics

Baseline characteristics of the 106 patients are summarized in Table 1. The cohort consisted of older adults (median age, 66 years; 63.2% men) with a high prevalence of cardiovascular risk factors (50.0% with hypertension and 32.1% with diabetes), and most PET/CT scans (82.1%) were performed for initial cancer staging.

The distribution of Agatston scores on reference ECG-gated chest CT showed a broad range. The median Agatston score was 47 (IQR, 0–388). During a median follow-up of approximately 3.5 years (maximum, 9 years), 14 patients (13.2%) experienced at least one MACE. Specifically, nine patients (8.5%) underwent coronary angiography followed by revascularization (stenting or bypass surgery) after PET/CT and ECG-gated CT. Two patients (1.9%) had ischemic stroke, and one patient each (0.9%) had myocardial infarction, acute coronary syndrome (unstable angina/non-ST elevation myocardial infarction without intervention), and new-onset complete atrioventricular block requiring pacemaker implantation. The remaining 92 patients (86.8%) did not experience MACE during follow-up.

### 3.2. Interobserver Agreement of Standard Visual and Ordinal Scales on PET/CT

Interobserver agreement for CAC grading using both the standard visual and ordinal scales on PET/CT was good (Table 3). For the standard visual scale, the overall kappa among the six readers was 0.761 (95% CI, 0.710–0.811); agreement was higher among radiologists (κ = 0.823) than among nuclear medicine physicians (κ = 0.681). The ordinal scale showed a similar overall kappa of 0.779 (95% CI, 0.728–0.829), with less difference between reader groups (radiologists, κ = 0.816; nuclear medicine physicians, κ = 0.781).

### 3.3. Concordance with Agatston Score Categories from ECG-Gated Chest CT

Agatston score measured on ECG-gated chest CT was significantly higher than on PET/CT (mean ± SD, 314 ± 613 vs. 128 ± 380; median [IQR], 47 [0–388] vs. 0 [0–75]; *p* < 0.001). Agreement between the Agatston score categories on ECG-gated chest CT and the three PET/CT-derived scales is summarized in Table 4. Among the three, the standard visual scale demonstrated the highest concordance (κ = 0.849; 95% CI: 0.709–0.989), followed by the ordinal scale (κ = 0.750; 95% CI: 0.616–0.885). Both the standard visual and ordinal scales more frequently underestimated CAC severity (17.0% and 27.4%, respectively) than overestimated it (1.9% for both), and both outperformed the PET/CT-derived Agatston categories (κ = 0.464; 95% CI: 0.348–0.580).

### 3.4. Prognostic Value for MACE

Kaplan–Meier curves showed progressively lower MACE-free survival with increasing CAC severity as assessed by PET/CT (Figure 3). This trend was statistically significant for the ordinal scale (*p* = 0.047) and was marginally significant for the standard visual scale (*p* = 0.084). In univariable Cox regression analysis, the hazard ratios for severe CAC compared with no CAC were 4.41 (95% CI: 1.09–17.81, *p* = 0.037) using the standard visual scale and 6.59 (95% CI: 1.27–34.29, *p* = 0.025) using the ordinal scale.

## 4. Discussion

Our study demonstrates that standard visual and ordinal CAC scoring on routine [^18^F]FDG PET/CT scans are reproducible, clinically relevant, and closely align with Agatston scoring on ECG-gated chest CT. Unlike prior studies that primarily involved cardiothoracic radiologists or specially trained readers [15,16], we uniquely assessed interobserver reproducibility among nuclear medicine physicians using only brief written instructions. Despite the limitations of free-breathing acquisition and thicker slices on PET/CT, both standard visual and ordinal scoring methods showed strong reproducibility and meaningful prognostic value.

Ungated PET/CT may be the first imaging study to reveal CAC in patients who do not undergo diagnostic chest CT, such as those with abdominal malignancies [25]. However, guidance for nuclear medicine physicians remains limited. The structured ordinal scale demonstrated good interobserver agreement (overall, κ = 0.779; radiologists, κ = 0.816; nuclear medicine physicians, κ = 0.781) and substantial concordance with ECG-gated chest CT Agatston score categories (κ = 0.750). The unstructured standard visual scale showed excellent agreement among radiologists (κ = 0.823) but lower agreement among nuclear medicine physicians (κ = 0.681), whereas the structured ordinal scale yielded more similar kappa values between reader groups. This disparity likely reflects differences in prior exposure to CAC scoring and the fact that our readers received only brief written instructions without interactive training or feedback. Notably, the structured ordinal scale achieved substantial agreement even among nuclear medicine physicians, suggesting that less-experienced readers may particularly benefit from simple, standardized guidance. In this context, stepwise instructions and representative examples such as those provided in Appendix A may serve as a practical template to facilitate implementation and improve reproducibility in routine practice. Our results therefore represent a ‘minimal training’ scenario that is pragmatic but may underestimate the performance that could be achieved with standardized education or calibration sessions. Future work should evaluate whether short case-based training modules can further improve interobserver agreement, particularly among nuclear medicine physicians.

Both standard visual and ordinal assessments on low-dose PET/CT correlated well with Agatston score categories on ECG-gated chest CT (κ = 0.849 and 0.750, respectively), indicating that qualitative evaluation can reliably approximate coronary calcium burden. Direct Agatston scoring on PET/CT was less accurate (κ = 0.464). The systematic underestimation of CAC on PET/CT is likely driven by technical differences between the low-dose CT component and dedicated ECG-gated coronary CT. In our protocol, low-dose PET/CT images were reconstructed with 4 mm slice thickness and 4 mm intervals during free breathing and with lower tube current, whereas ECG-gated coronary CT used 2.5–3 mm slices, overlapping reconstruction, and breath-held acquisition. The coarser spatial resolution and respiratory and cardiac motion on PET/CT increase partial-volume averaging and image noise, causing small or low-density calcifications to fall below the Agatston threshold or to be visually inapparent [20]. These technical limitations should be considered when interpreting absent or mild CAC on PET/CT. In addition, semi-automated Agatston scoring requires time-consuming manual region-of-interest placement and is not routinely supported by PET/CT workstations. In clinical practice, standard visual and ordinal CAC scoring can be performed rapidly during routine PET/CT interpretation without additional software, making them attractive for busy oncology workflows.

Although standard visual and ordinal scoring on low-dose PET/CT showed good agreement with Agatston score categories on ECG-gated chest CT, discordance between PET/CT and ECG-gated CT predominantly reflected underestimation in the lower CAC categories. On the standard visual PET/CT scale, 12 of 30 patients (40.0%) with mild CAC on ECG-gated CT were categorized as having no CAC, whereas only 5 of 21 (23.8%) patients with moderate CAC and 1 of 26 (3.8%) with severe CAC were underestimated by one category or more. Overestimation was rare across all categories. These discordance patterns were also reported in the study by Fresno et al. [16] that applied ordinal scoring to ungated chest CT. Visual analysis of ungated chest CT may miss or underestimate mild CAC, whereas moderate-to-severe CAC is generally preserved even with thicker slices and motion artifacts. Clinically, this suggests that a PET/CT report describing “no CAC” does not rule out mild coronary calcification, whereas the presence of moderate-to-severe CAC on PET/CT is indicative of a high Agatston burden on ECG-gated CT and may warrant more aggressive cardiovascular risk assessment.

Importantly, CAC severity assessed via standard visual and ordinal scales provided prognostic information. Patients with severe CAC had significantly higher rates of MACE during follow-up, and Kaplan–Meier curves showed clear stratification by CAC category. These results are consistent with established evidence of the prognostic value of CAC in asymptomatic populations [12] and recent oncology data showing that high CAC burden on baseline PET/CT is associated with increased mortality and cardiovascular events [25]. Although our cohort was not sufficiently powered for multivariable analysis, the observed associations suggest that CAC on PET/CT is associated with an increased risk of MACE; however, whether this association is independent of traditional cardiovascular risk factors remains to be determined in larger cohorts.

From a clinical perspective, routine reporting of incidental CAC could improve cardiovascular risk stratification in patients with cancer, who often face additional perioperative and treatment-related cardiovascular risks. The current SCCT/STR guidelines recommend visual assessment of CAC on all ungated chest CT scans [14]. However, underreporting persists, in part because nuclear medicine physicians may be uncertain about the clinical implications of CAC. Severe CAC should prompt consideration of cardiology referral, intensified risk factor management, and possibly further evaluation, such as stress myocardial perfusion imaging or coronary angiography [26,27]. Two main barriers to consistent reporting include limited familiarity with CAC’s prognostic value and lack of awareness among nuclear medicine physicians. Our findings address both concerns and support incorporating standard visual and ordinal scoring into routine PET/CT interpretation.

Recent studies have expanded coronary risk assessment beyond traditional Agatston categories, incorporating visual ordinal scores, risk-stratified reporting systems, and automated or deep-learning-based quantification of CAC on non-gated CT. These approaches, summarized in a recent comprehensive review [28], highlight the potential of opportunistic CAC assessment across a wide variety of CT protocols. Our findings complement this literature by demonstrating that simple qualitative scales applied to low-dose PET/CT data, even by nuclear medicine physicians with minimal training, can reproduce ECG-gated Agatston categories and provide preliminary prognostic information. In settings where advanced software or artificial-intelligence tools are not yet available, such structured visual approaches may offer an immediately implementable strategy to improve cardiovascular risk stratification.

This study has several limitations. First, its retrospective, single-center design and the requirement that patients undergo both PET/CT and ECG-gated chest CT within 90 days introduced selection bias toward individuals with a higher clinical suspicion for coronary artery disease; therefore, our findings may not be directly generalizable to the broader population of patients undergoing oncologic PET/CT. Second, the modest sample size and limited number of MACE precluded adequately powered multivariable adjustment for traditional cardiovascular risk factors, so we could only demonstrate associations in univariable analyses and cannot claim independence from established risk markers. Additionally, only 14 MACE occurred during a median follow-up of 3.5 years, resulting in wide confidence intervals around the hazard ratio estimates; thus, the magnitude of the association between CAC severity and MACE should be interpreted with caution. Third, the ungated, low-dose CT component of PET/CT used thicker slices, free-breathing acquisition, and lower tube current than dedicated ECG-gated coronary CT, and these technical differences likely contributed to the systematic underestimation of CAC. Fourth, we did not systematically collect information on cardiotoxic cancer therapies administered between imaging and clinical follow-up, which may have influenced cardiovascular outcomes. Finally, although all readers were provided with written instructions for CAC scoring, the lower agreement among nuclear medicine physicians suggests that additional standardized training and case-based feedback may be necessary for optimal implementation in routine practice. Nevertheless, the use of paired imaging, standardized scoring schemes, and blinded assessments by multiple independent readers provides robust internal validity.

## 5. Conclusions

Standard visual and ordinal CAC scoring on routine PET/CT is feasible, reproducible, and prognostically informative. Integrating qualitative CAC scoring into routine PET/CT interpretation could enhance cardiovascular risk stratification and management in patients with cancer, potentially improving long-term outcomes.

## Figures and Tables

**Figure 1 diagnostics-15-02969-f001:**
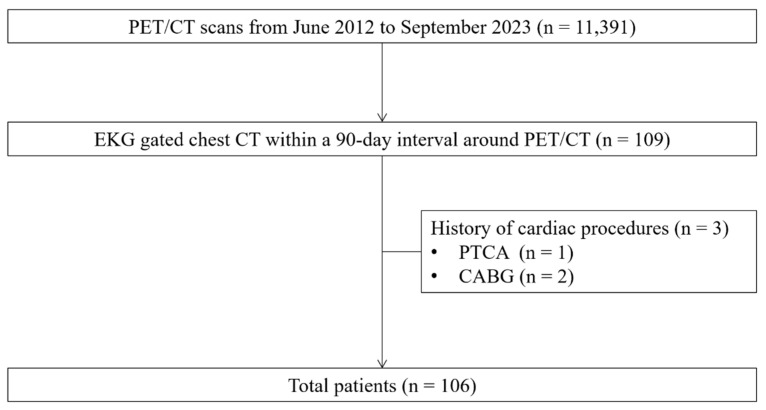
Patient flow diagram.

**Figure 2 diagnostics-15-02969-f002:**
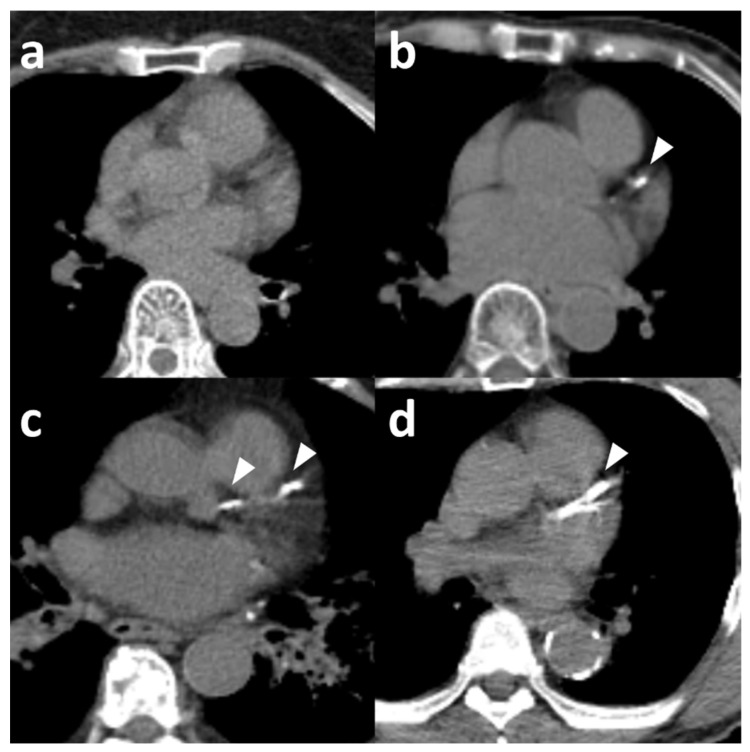
Representative examples of none (**a**), mild (**b**), moderate (**c**) and severe (**d**) coronary artery calcification (arrow head) on low-dose CT images from PET/CT.

**Figure 3 diagnostics-15-02969-f003:**
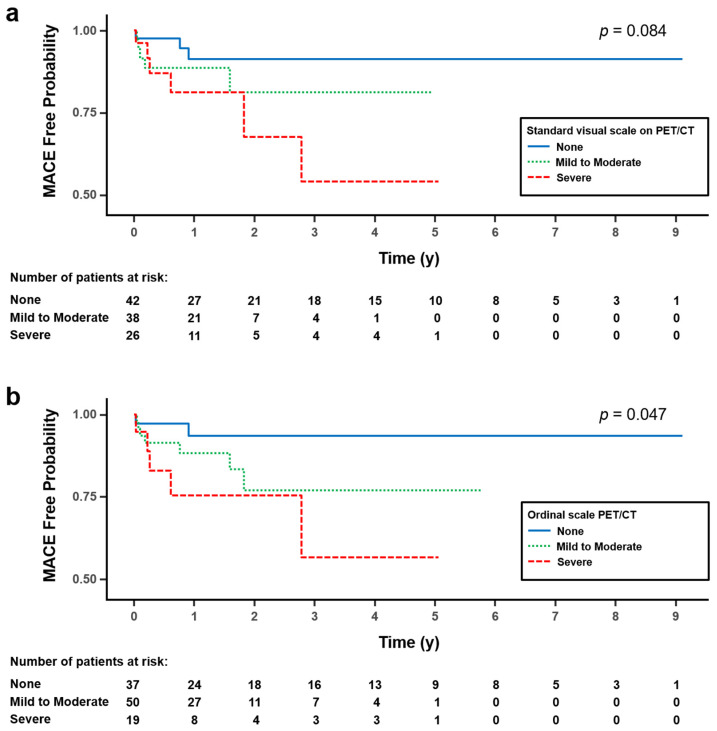
Kaplan–Meier survival curves for major adverse cardiac events (MACE) according to coronary artery calcification (CAC) severity on PET/CT. A progressively lower MACE-free survival trend with increasing CAC severity was marginally significant for the standard visual scale ((**a**), *p* = 0.084) and statistically significant for the ordinal scale ((**b**), *p* = 0.047). In both the standard visual (**a**) and ordinal (**b**) scales, patients with severe CAC (green curves) had significantly lower MACE-free survival (*p* = 0.037 and 0.025, respectively) compared with those with no CAC (red curves).

**Table 1 diagnostics-15-02969-t001:** Patient characteristics.

Characteristics	*n* = 106
Age, year, median [IQR]	66 [60–75]
Sex	
Male, *n* (%)	67 (63.2%)
Female, *n* (%)	39 (36.8%)
Risk factor	
Hypertension, *n* (%)	53 (50.0%)
Diabetes mellitus, *n* (%)	34 (32.1%)
Smoking history, *n* (%)	30 (28.3%)
Dyslipidemia, *n* (%)	23 (21.7%)
Medication	
Statin, *n* (%)	47 (44.3%)
ACEi/ARB, *n* (%)	39 (36.8%)
Calcium channel blocker, *n* (%)	30 (28.3%)
Antiplatelet agent, *n* (%)	22 (20.8%)
β-blocker, *n* (%)	11 (10.4%)
Purpose of PET/CT scans	
Cancer staging, *n* (%)	87 (82.1%)
Cancer recurrence evaluation, *n* (%)	11 (10.4%)
Healthcare, *n* (%)	4 (3.8%)
Cancer response evaluation, *n* (%)	3 (2.8%)
Infection, *n* (%)	1 (0.9%)
Underlying malignancy	
Colorectal cancer, *n* (%)	18 (17.0%)
Lung cancer, *n* (%)	16 (15.1%)
Stomach cancer, *n* (%)	12 (11.3%)
Head and neck cancer, *n* (%)	12 (11.3%)
Breast cancer, *n* (%)	8 (7.5%)
Cervical cancer, *n* (%)	5 (4.7%)
Thyroid cancer, *n* (%)	5 (4.7%)
Cholangiocarcinoma, *n* (%)	4 (3.8%)
Endometrial cancer, *n* (%)	3 (2.8%)
Pancreatic cancer, *n* (%)	3 (2.8%)
Prostate cancer, *n* (%)	2 (1.9%)
Other, *n* (%)	6 (5.7%)
None, *n* (%)	12 (11.3%)
Stage of malignancy	
Limited stage, *n* (%)	48 (45.3%)
Advanced stage, *n* (%)	46 (43.4%)
None, *n* (%)	12 (11.3%)
Interval between PET/CT and ECG-gated CT, median [IQR]	12 [4–28]
CAC score on ECG-gated chest CT, median [IQR]	47 [0–388]
None (score, 0), *n* (%)	29 (27.4%)
Mild (score, 1–100), *n* (%)	30 (28.3%)
Moderate (score, 101–400), *n* (%)	21 (19.8%)
Severe (score, >400), *n* (%)	26 (24.5%)
Major adverse cardiovascular events	
Coronary angiogram followed by revascularization, *n* (%)	9 (8.5%)
Stroke, *n* (%)	2 (1.9%)
Myocardial infarction, *n* (%)	1 (0.9%)
Acute coronary syndrome, *n* (%)	1 (0.9%)
Complete atrioventricular block, *n* (%)	1 (0.9%)
None, *n* (%)	92 (86.8%)

ACEi, angiotensin-converting enzyme inhibitor; ARB, angiotensin receptor blocker; CAC, coronary artery calcium; CT, computed tomography; IQR, interquartile range; PET, positron emission tomography.

**Table 2 diagnostics-15-02969-t002:** Imaging acquisition.

	PET/CT (*n* = 106)	CT 1 (*n* = 77)	CT 2 (*n* = 18)	CT 3 (*n* = 11)
Model	Gemini TF 16	SOMATOM Definition Flash	SOMATOM Drive	Revolution Apex
Manufacturer	Philips Healthcare	Siemens Healthineers	Siemens Healthineers	GE Healthcare
Tube voltage	120 kVp	120 kVp	100 kVp *	120 kVp
Tube current	50 mAs	60 mAs	120 mAs	80 mAs
Mode	Helical	Step and shoot	Step and shoot	Axial volume
Pitch	0.625	N/A	N/A	N/A
Slice thickness	4 mm	3 mm	3 mm	2.5 mm
Interval	4 mm	1.5 mm	1.5 mm	2.5 mm
Collimation	8 × 3.0 mm	64 × 0.6 mm	64 × 0.6 mm	256 × 0.625 mm
ECG gating	No	Yes	Yes	Yes
Breathing	free	hold	hold	hold

* 100 kVp data were reconstructed using Siemens’ kV-independent CaScoring kernel (Sa36), resulting in HU values equivalent to 120 kVp images for Agatston scoring. CT, computed tomography; ECG, electrocardiogram; N/A, not applicable; PET, positron emission tomography.

**Table 3 diagnostics-15-02969-t003:** Interobserver agreement of standard visual and ordinal scales on PET/CT.

	Radiologists (*n* = 3)	Nuclear Medicine Physicians (*n* = 3)	All Physicians (*n* = 6)
Standard visual scale	0.823 [0.773–0.873]	0.681 [0.606–0.756]	0.761 [0.710–0.811]
Ordinal scale	0.816 [0.761–0.870]	0.781 [0.720–0.843]	0.779 [0.728–0.829]

Data are presented as kappa values (95% confidence intervals).

**Table 4 diagnostics-15-02969-t004:** Agreement between Agatston score categories on ECG-gated chest CT and various scales on PET/CT.

Scale on PET/CT	Agatston Score Categories on ECG-Gated Chest CT				*k* [95% CI]
	None (0)	Mild (1–100)	Moderate (101–400)	Severe (>400)	
Agatston score categories on PET/CT					0.464 [0.348–0.580]
None (0)	29 (27.4%)	22 (20.8%)	6 (5.7%)	0 (0.0%)	
Mild (1–100)	0 (0.0%)	8 (7.5%)	13 (12.3%)	7 (6.6%)	
Moderate (101–400)	0 (0.0%)	0 (0.0%)	2 (1.9%)	11 (10.4%)	
Severe (>400)	0 (0.0%)	0 (0.0%)	0 (0.0%)	8 (7.5%)	
Standard visual scale on PET/CT					0.849 [0.709–0.989]
None	29 (27.4%)	12 (11.3%)	1 (0.9%)	0 (0.0%)	
Mild	0 (0.0%)	17 (16.0%)	4 (3.8%)	0 (0.0%)	
Moderate	0 (0.0%)	1 (0.9%)	15 (14.2%)	1 (0.9%)	
Severe	0 (0.0%)	0 (0.0%)	1 (0.9%)	25 (23.6%)	
Ordinal scale on PET/CT					0.750 [0.616–0.885]
None	28 (26.4%)	8 (7.5%)	1 (0.9%)	0 (0.0%)	
Mild	1 (0.9%)	22 (20.8%)	12 (11.3%)	1 (0.9%)	
Moderate	0 (0.0%)	0 (0.0%)	7 (6.6%)	7 (6.6%)	
Severe	0 (0.0%)	0 (0.0%)	1 (0.9%)	18 (17.0%)	

Data are presented as *n* (%) or kappa values [95% confidence intervals]. CI, confidence interval; CT, computed tomography; PET, positron emission tomography.

## Data Availability

The raw data supporting the conclusions of this article will be made available by the authors on request. The data are not publicly available due to privacy.

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
