# Peer review of "Standard Visual and Ordinal Coronary Calcium Scoring on PET/CT: Agreement with Agatston Scoring and Prognostic Implications"

_diagnostics, 2025, doi:10.3390/diagnostics15232969_

Round 1
Reviewer 1 Report
Comments and Suggestions for Authors
Dear authors,
Thank you very much for submitting your paper regarding Coronary Calcium Scoring (CAC) on PET/CT, its agreement with Agaston scoring and the relevant prognostic implications. This was an interesing retrospective study evaluating coronary vessel calcification on routine PET/CT studies by means of visual, ordinal and Agaston scoring methods. Comparison with ECG-gated CT (standard of truth) was performed. The study demonstrated good interobserver agreement and reasonable concordance with results of ECG-gated CT, as well as an association of PET/CT CAC scoring with future major adverse CVD events. Below, there areas that may need further clarification, questions to be addressed and suggestions for improvement:
- The main limitation of the study is the retrospective nature (inherent limitation) and the fact that inclusion criteria required both PET/CT and ECG-gated CT within a resonable time frame. This introduces selection bias, since this is not a random sample. Patients undergoing ECG-gated CT most likely represent a high-risk population with suspected CHD. Thus, results may not be generalized to routine oncologic PET/CT pts. Please elaborate this limitation. Also, ensure that all major limitations of this study are presented in detail in the last paragraph. The current limitations paragraph is too brief.
- lines 187-189 should be removed (remnants of the pre-formatted text not relevant with the paper)
- lines 191-199 are repetition of the table results, may be removed. Most pertinent demographic and clinical characteristics may be concisely summarized in one-two sentences, with full details left in the table.
- I think results of Agaston scores on ECG-gated chest CT should be presented in detail with relevant statistics (mean, SDs, range etc), since this is the standard of truth method. How do these compare to general population or other high risk populations ?
- There is no multivariate analysis performed. Why not ? Without adjusting for traditional CVD risk factors (e.g. hypertension, DM, dyslipidaemia), the independent prognostic value of PET/CT scoring remains uncertain. This limitation should be clearly stated. I do appreciate the sample-size is modest for such analysis.
- Nuclear medicine physicians showed lower agreement scores compared with radiologists. Could this be due to the limited training provided ? The use of only written instructions may represent a significant limitation, potentially affecting the validity of the results. Please discuss how reader training and experience may have affected scoring performance.
- Another important methodological consideration is that there is no analysis of the discordance (and relevant treands) between PET/CT and ECG-gated CAC scoring. Having a look on the table, it seems that PET/CT, systematically, underestimates CAC. For example on visual scale: 12/29 patients with mild CAC on gated CT were missed and classified as "none". This substantial underestimation has significant clinical implications that are underexplored and should be analyzed.
- Technical limitations of PET play a role for previous limitation. Lower (4 mm) slice thickness on PET/CT than gated CT (2.5–3 mm), free-breathing acquisition, lower current, substantial partial volume averaging, noise are factors leading to PET/CT missing small calcifications. Please provide specific discussion of how these technical factors may contribute to underestimation.
Author Response
Dear reviewers: We deeply appreciate the reviewers’ insightful and constructive comments on our manuscript entitled “Standard Visual and Ordinal Coronary Calcium Scoring on PET/CT: Agreement with Agatston Scoring and Prognostic Implications (manuscript number: diagnostics-3975098)”, which has afforded our manuscript the opportunity for further refinement. We have made every effort to address these concerns and have revised the manuscript accordingly. Our point-by-point responses to the reviewers' comments and the corresponding corrections in the manuscript are highlighted in blue. We hope that the revised manuscript will be accepted for publication in Diagnostics.
Reviewer #1:
Thank you very much for submitting your paper regarding Coronary Calcium Scoring (CAC) on PET/CT, its agreement with Agaston scoring and the relevant prognostic implications. This was an interesting retrospective study evaluating coronary vessel calcification on routine PET/CT studies by means of visual, ordinal and Agaston scoring methods. Comparison with ECG-gated CT (standard of truth) was performed. The study demonstrated good interobserver agreement and reasonable concordance with results of ECG-gated CT, as well as an association of PET/CT CAC scoring with future major adverse CVD events. Below, there areas that may need further clarification, questions to be addressed and suggestions for improvement:
• The main limitation of the study is the retrospective nature (inherent limitation) and the fact that inclusion criteria required both PET/CT and ECG-gated CT within a reasonable time frame. This introduces selection bias, since this is not a random sample. Patients undergoing ECG-gated CT most likely represent a high-risk population with suspected CHD. Thus, results may not be generalized to routine oncologic PET/CT pts. Please elaborate this limitation. Also, ensure that all major limitations of this study are presented in detail in the last paragraph. The current limitations paragraph is too brief.
→ Thank you for the valuable comment. We fully agree that the retrospective design and inclusion of only patients who underwent both PET/CT and ECG-gated chest CT introduce selection bias and limit generalizability. We have now expanded the limitations as follows Ex) Line 337: This study has several limitations. First, its retrospective, single-center design and the requirement that patients undergo both PET/CT and ECG-gated chest CT within 90 days introduced selection bias toward individuals with a higher clinical suspicion for coronary artery disease; therefore, our findings may not be directly generalizable to the broader population of patients undergoing oncologic PET/CT. Second, the modest sample size and limited number of MACE events precluded adequately powered multivariable adjustment for traditional cardiovascular risk factors, so we could only demonstrate associations in univariable analyses and cannot claim independence from established risk markers. Additionally, only 14 MACE events occurred during a median follow-up of 3.5 years, resulting in wide confidence intervals around the hazard ratio estimates; thus, the magnitude of the association between CAC severity and MACE should be interpreted with caution. Third, the ungated, low-dose CT component of PET/CT used thicker slices, free-breathing acquisition, and lower tube current than dedicated ECG-gated coronary CT, and these technical differences likely contributed to the systematic underestimation of CAC. Fourth, we did not systematically collect information on cardiotoxic cancer therapies administered between imaging and clinical follow-up, which may have influenced cardiovascular outcomes. Finally, although all readers were provided with written instructions for CAC scoring, the lower agreement among nuclear medicine physicians suggests that additional standardized training and case-based feedback may be necessary for optimal implementation in routine practice. Nevertheless, the use of paired imaging, standardized scoring schemes, and blinded assessments by multiple independent readers provides robust internal validity.
• lines 187-189 should be removed (remnants of the pre-formatted text not relevant with the paper)
→ Thank you for your thorough check. We have removed the residual template sentence.
• lines 191-199 are repetition of the table results, may be removed. Most pertinent demographic and clinical characteristics may be concisely summarized in one-two sentences, with full details left in the table.
→ We have shortened the text describing baseline characteristics and now summarize them more concisely. Ex) Line 189: Baseline characteristics of the 106 patients are summarized in Table 1. The cohort consisted of older adults (median age, 66 years; 63.2% men) with a high prevalence of cardiovascular risk factors (50.0% with hypertension and 32.1% with diabetes), and most PET/CT scans (82.1%) were performed for initial cancer staging.
• I think results of Agaston scores on ECG-gated chest CT should be presented in detail with relevant statistics (mean, SDs, range etc.), since this is the standard of truth method. How do these compare to general population or other high risk populations?
→ We appreciate this comment. We agree that the reference ECG-gated CT Agatston scores should be more fully characterized. We added the following sentence to the result. Ex) Line 213: Agatston score measured on ECG-gated chest CT was significantly higher than on PET/CT (mean±SD, 314±613 vs. 128±380; median [IQR], 47 [0–388] vs. 0 [0–75]; p < 0.001).
• There is no multivariate analysis performed. Why not? Without adjusting for traditional CVD risk factors (e.g. hypertension, DM, dyslipidaemia), the independent prognostic value of PET/CT scoring remains uncertain. This limitation should be clearly stated. I do appreciate the sample-size is modest for such analysis.
→ We fully agree. We originally refrained from multivariable Cox regression because the number of MACE events (n = 14) was too small to allow a stable model including several cardiovascular risk factors, and we wanted to avoid overfitting. To address your comment, we have now explicitly stated this limitation and removed any implication of “independent” prognostic value. Ex) Line 310: Although our cohort was not sufficiently powered for multivariable analysis, the observed associations suggest that CAC on PET/CT is associated with an increased risk of MACE; however, whether this association is independent of traditional cardiovascular risk factors remains to be determined in larger cohorts. Ex) Line 341: Second, the modest sample size and limited number of MACE events precluded adequately powered multivariable adjustment for traditional cardiovascular risk factors, so we could only demonstrate associations in univariable analyses and cannot claim independence from established risk markers.
• Nuclear medicine physicians showed lower agreement scores compared with radiologists. Could this be due to the limited training provided? The use of only written instructions may represent a significant limitation, potentially affecting the validity of the results. Please discuss how reader training and experience may have affected scoring performance.
→ Thank you for highlighting this important point. We agree that differences in training and experience are a likely explanation for the lower kappa values among nuclear medicine physicians. We have expanded the Discussion to address this explicitly. Ex) Line 258: The unstructured standard visual scale showed excellent agreement among radiologists (κ = 0.823) but lower agreement among nuclear medicine physicians (κ = 0.681), whereas the structured ordinal scale yielded more similar kappa values between reader groups. This disparity likely reflects differences in prior exposure to CAC scoring and the fact that our readers received only brief written instructions without interactive training or feedback. Notably, the structured ordinal scale achieved substantial agreement even among nuclear medicine physicians, suggesting that less-experienced readers may particularly benefit from simple, standardized guidance. In this context, stepwise instructions and representative examples such as those provided in Supplementary Material 2 may serve as a practical template to facilitate implementation and improve re-producibility in routine practice. Our results therefore represent a ‘minimal training’ scenario that is pragmatic but may underestimate the performance that could be achieved with standardized education or calibration sessions. Future work should evaluate whether short case-based training modules can further improve interobserver agreement, particularly among nuclear medicine physicians.
• Another important methodological consideration is that there is no analysis of the discordance (and relevant trends) between PET/CT and ECG-gated CAC scoring. Having a look on the table, it seems that PET/CT, systematically, underestimates CAC. For example on visual scale: 12/29 patients with mild CAC on gated CT were missed and classified as "none". This substantial underestimation has significant clinical implications that are underexplored and should be analyzed.
→ We appreciate this insightful suggestion. We agree that a more explicit analysis of discordance patterns is important and clinically relevant. These discordance patterns were also observed in the study by Fresno et al., which applied visual ordinal scoring to ungated chest CT. We have added the following paragraph to the Discussion section. Ex) Line 290: Although standard visual and ordinal scoring on low-dose PET/CT showed good agreement with Agatston score categories on ECG-gated chest CT, discordance be-tween PET/CT and ECG-gated CT predominantly reflected underestimation in the lower CAC categories. On the standard visual PET/CT scale, 12 of 30 patients (40.0%) with mild CAC on ECG-gated CT were categorized as having no CAC, whereas only 5 of 21 (23.8%) patients with moderate CAC and 1 of 26 (3.8%) with severe CAC were underestimated by one category or more. Overestimation was rare across all categories. These discordance patterns were also reported in the study by Fresno et al. [16] that applied ordinal scoring to ungated chest CT. Visual analysis of ungated chest CT may miss or underestimate mild CAC, whereas moderate-to-severe CAC is generally pre-served even with thicker slices and motion artifacts. Clinically, this suggests that a PET/CT report describing “no CAC” does not rule out mild coronary calcification, whereas the presence of moderate-to-severe CAC on PET/CT is indicative of a high Agatston burden on ECG-gated CT and may warrant more aggressive cardiovascular risk assessment.
• Technical limitations of PET play a role for previous limitation. Lower (4 mm) slice thickness on PET/CT than gated CT (2.5–3 mm), free-breathing acquisition, lower current, substantial partial volume averaging, noise are factors leading to PET/CT missing small calcifications. Please provide specific discussion of how these technical factors may contribute to underestimation.
→ We fully agree and have extended the Discussion to provide a more explicit link between technical factors and underestimation. Ex) Line 276: The systematic underestimation of CAC on PET/CT is likely driven by technical differences between the low-dose CT component and dedicated ECG-gated coronary CT. In our protocol, low-dose PET/CT images were reconstructed with 4-mm slice thickness and 4-mm intervals during free breathing and with lower tube current, whereas ECG-gated coronary CT used 2.5–3-mm slices, overlapping reconstruction, and breath-held acquisition. The coarser spatial resolution and respiratory and cardiac motion on PET/CT increase partial-volume averaging and image noise, causing small or low-density calcifications to fall below the Agatston threshold or to be visually inapparent [20]. These technical limitations should be considered when interpreting absent or mild CAC on PET/CT.
Reviewer 2 Report
Comments and Suggestions for Authors
The inclusion criteria create significant selection bias. Patients who underwent both PET/CT AND ECG-gated chest CT within 90 days represent a highly selected population - likely enriched for cardiovascular concerns. This limits generalizability to the typical oncology PET/CT population. The authors acknowledge this briefly but don't adequately address how this affects their conclusions about routine implementation.
With only 14 MACE events over 3.5 years in 106 patients, the study is severely underpowered for survival analysis. The Cox regression results (HR 4.41 and 6.59) have very wide confidence intervals (1.09-17.81 and 1.27-34.29), indicating substantial uncertainty. Therefore, I suggest acknowledging this as an important limitaton. Also, I suggest to avoid strong claims about prognostic value, given limited events
The authors acknowledge (line 285-286) that the cohort wasn't powered for multivariable analysis, yet they claim CAC "may serve as an independent risk factor." Either the authors should perform multivariable analysis or remove claims about "independent" prognostic value.
Which software version was used for scoring? Who performed the Agatston scores on PET/CT and gated CT? Were they blinded?
A 90-day window between PET/CT and gated CT is quite wide. CAC can progress, especially in high-risk patients. Providing median time between scans and justification for the 90-day cutoff would increase the quality of the manuscirpt.
Many patients (43.4% advanced stage) likely received chemotherapy or radiation between or after scans. Some treatments affect cardiovascular risk. Did the authors collected data on cardiotoxic therapies?
The Discussion states this method is quick, reproducible without specialized training. I have not seen data on actual reading time provided, and no data on the feasibility of implementation in routine practice. Also, there is a missing comparison with emerging CAC metrics and contemporary evidence. The manuscript focuses solely on traditional CAC scoring methods (Agatston categories and visual scales) but does not discuss or compare findings with recent advances in coronary risk assessment from non-gated CT. I recommend the authors review and incorporate relevant findings. If considered useful, they can use an elegant, comprehensive ( DOI:10.3390/jcm13175205) review for this section.
Author Response
Dear reviewers: We deeply appreciate the reviewers’ insightful and constructive comments on our manuscript entitled “Standard Visual and Ordinal Coronary Calcium Scoring on PET/CT: Agreement with Agatston Scoring and Prognostic Implications (manuscript number: diagnostics-3975098)”, which has afforded our manuscript the opportunity for further refinement. We have made every effort to address these concerns and have revised the manuscript accordingly. Our point-by-point responses to the reviewers' comments and the corresponding corrections in the manuscript are highlighted in blue. We hope that the revised manuscript will be accepted for publication in Diagnostics.
Reviewer #2:
The inclusion criteria create significant selection bias. Patients who underwent both PET/CT AND ECG-gated chest CT within 90 days represent a highly selected population - likely enriched for cardiovascular concerns. This limits generalizability to the typical oncology PET/CT population. The authors acknowledge this briefly but don't adequately address how this affects their conclusions about routine implementation.
→ Thank you for this important clarification. As also noted by Reviewer #1, we agree that our cohort represents a selected, higher-risk group and that this limits generalizability. We modified limitations as follows. Ex) Line 337: First, its retrospective, single-center design and the requirement that patients undergo both PET/CT and ECG-gated chest CT within 90 days introduced selection bias toward individuals with a higher clinical suspicion for coronary artery disease; therefore, our findings may not be directly generalizable to the broader population of patients undergoing oncologic PET/CT. With only 14 MACE events over 3.5 years in 106 patients, the study is severely underpowered for survival analysis. The Cox regression results (HR 4.41 and 6.59) have very wide confidence intervals (1.09-17.81 and 1.27-34.29), indicating substantial uncertainty. Therefore, I suggest acknowledging this as an important limitation. Also, I suggest to avoid strong claims about prognostic value, given limited events → We agree and appreciate this comment. Ex) Line 345: Additionally, only 14 MACE events occurred during a median follow-up of 3.5 years, resulting in wide confidence intervals around the hazard ratio estimates; thus, the magnitude of the association between CAC severity and MACE should be interpreted with caution.
The authors acknowledge (line 285-286) that the cohort wasn't powered for multivariable analysis, yet they claim CAC "may serve as an independent risk factor." Either the authors should perform multivariable analysis or remove claims about "independent" prognostic value.
→ We fully agree. As noted in our response to Reviewer #1, we have chosen not to perform multivariable Cox regression because the event count is insufficient for stable models. To address this, we have now explicitly stated this limitation and removed any implication of “independent” prognostic value. Ex) Line 310: Although our cohort was not sufficiently powered for multivariable analysis, the observed associations suggest that CAC on PET/CT is associated with an increased risk of MACE; however, whether this association is independent of traditional cardiovascular risk factors remains to be determined in larger cohorts. Ex) Line 341: Second, the modest sample size and limited number of MACE events precluded adequately powered multivariable adjustment for traditional cardiovascular risk factors, so we could only demonstrate associations in univariable analyses and cannot claim independence from established risk markers.
Which software version was used for scoring? Who performed the Agatston scores on PET/CT and gated CT? Were they blinded?
→ Thank you for pointing out the need for clarification. Ex) Line 144: Agatston scores were obtained from both the ECG-gated chest CT and the ungated PET/CT using dedicated software (SmartScore 4.0; GE Healthcare, USA) on an advanced workstation (Advantage Workstation 4.7; GE Healthcare, USA) by the corresponding author, who was blinded to any clinical information during the analysis.
A 90-day window between PET/CT and gated CT is quite wide. CAC can progress, especially in high-risk patients. Providing median time between scans and justification for the 90-day cutoff would increase the quality of the manuscript.
→ Thank you for your valuable comments. We have added the median time and interquartile range to Table 1. Fresno et al. (doi: 10.2214/ajr.22.27664) used a 1-year window between ECG-gated CT and ungated CT; however, we considered this interval too wide and therefore shortened the range to 90 days to establish a more reasonable time frame.
Many patients (43.4% advanced stage) likely received chemotherapy or radiation between or after scans. Some treatments affect cardiovascular risk. Did the authors collected data on cardiotoxic therapies?
→ This is an excellent point. We did not systematically collect detailed data on specific cardiotoxic cancer therapies administered between imaging and follow-up. We now acknowledge this explicitly as a limitation. Ex) Line 351: Fourth, we did not systematically collect information on cardiotoxic cancer therapies administered between imaging and clinical follow-up, which may have influenced cardiovascular outcomes.
The Discussion states this method is quick, reproducible without specialized training. I have not seen data on actual reading time provided, and no data on the feasibility of implementation in routine practice. Also, there is a missing comparison with emerging CAC metrics and contemporary evidence. The manuscript focuses solely on traditional CAC scoring methods (Agatston categories and visual scales) but does not discuss or compare findings with recent advances in coronary risk assessment from non-gated CT. I recommend the authors review and incorporate relevant findings. If considered useful, they can use an elegant, comprehensive (DOI:10.3390/jcm13175205) review for this section.
→ We appreciate these suggestions and have revised the Discussion accordingly. In the original manuscript we stated that standard visual and ordinal scoring take “less than 20 s and 40 s, respectively.” Because we did not formally record reading times, we have removed these quantitative estimates and now describe feasibility more qualitatively. Ex) Line 287: In clinical practice, standard visual and ordinal CAC scoring can be performed rapidly during routine PET/CT interpretation without additional software, making them attractive for busy oncology workflows. → We also toned down “without specialized training” as discussed above and clarified that our setting reflects a minimal-training scenario rather than definitive evidence that no training is needed. Addition of contemporary evidence and emerging CAC metrics, we have added a new paragraph in the Discussion to place our results in the context of emerging CAC assessment methods from ungated CT, including deep-learning–based approaches. We have cited the suggested review as a new reference in the revised manuscript as follows. Ex) Line 326: Recent studies have expanded coronary risk assessment beyond traditional Agatston categories, incorporating visual ordinal scores, risk-stratified reporting systems, and automated or deep-learning-based quantification of CAC on non-gated CT. These approaches, summarized in a recent comprehensive review [28], highlight the potential of opportunistic CAC assessment across a wide variety of CT protocols. Our findings complement this literature by demonstrating that simple qualitative scales applied to low-dose PET/CT CT data, even by nuclear medicine physicians with minimal training, can reproduce ECG-gated Agatston categories and provide preliminary prognostic information. In settings where advanced software or artificial-intelligence tools are not yet available, such structured visual approaches may offer an immediately implementable strategy to improve cardiovascular risk stratification
Round 2
Reviewer 1 Report
Comments and Suggestions for Authors
Dear authors,
Thank you very much for all the effort to address all previous queries and remarks.
The limitations of the study have been properly highlighted, the discussion section was expanded and the overall quality of the paper was improved. I have no other comment to make or other suggestion for changes. Well done.
Reviewer 2 Report
Comments and Suggestions for Authors
The authors addressed all my comments, and I congratulate on their work!